# Acute febrile illness surveillance using TaqMan Array Cards in two urban health facilities, Monrovia, Liberia, December 2018–March 2020

Terrence Q. Lo[1]*, Elijah Paa Edu-Quansah[2], John Dogba[3], Fahn Taweh[3], Lekilay Tehmeh[4], Thomas Nagbe[3], Paul Whesseh[4], Dore Diabe[4], Eric Houpt[5], Jie Liu[5,6], Darwin J. Operario[5], Maame Amo-Addae[2], Davis Ashaba[2], Victoria Katawera[7], Daniel W. Martin[1], Denise Roth Allen[8], Amanda Balish[1], Barry Fields[1], Gulu Gwesa[8], Mosoka Fallah[3], Desmond Williams[8]

**1** US Centers for Disease Control and Prevention, Center for Global Health, Division of Global Health Protection, Atlanta, Georgia, United States of America, **2** African Field Epidemiology Network, Monrovia, Liberia, **3** National Public Health Institute of Liberia, Monrovia, Liberia, **4** Ministry of Health Liberia, Monrovia, Liberia, **5** Division of Infectious Diseases and International Health, University of Virginia, Charlottesville, Virginia, United States of America, **6** School for Public Health, Qingdao University, Qingdao, China, **7** College of Medicine and Veterinary Medicine, The University of Edinburgh, Edinburgh, Scotland, **8** US Centers for Disease Control and Prevention, Center for Global Health, Division of Global Health Protection, Monrovia, Liberia

☯ These authors contributed equally to this work.
* TGL8@cdc.gov

## Abstract

### Introduction

Fever is a common symptom of infectious diseases including for those with epidemic potential. Beyond malaria, the causes of undifferentiated (i.e., non-respiratory, non-diarrheal) acute febrile illnesses are not well characterized in Liberia.

### Methods

From December 2018 through March 2020, we established two acute febrile illness (AFI) sentinel surveillance sites in urban Monrovia at Redemption Hospital and Star of the Sea Health Centre, health facilities that were among the first to have Ebola cases during the 2014–2015 West Africa epidemic. Enrolled AFI patients were two (2) years of age or greater, had a measured fever of ≥37.5°C or history of fever within the past week, and without a known cause of fever. A standardized survey was administered to collect demographic, clinical characteristics, and risk factors. Whole blood was taken, nucleic acid material was extracted and ran on TaqMan Array Cards (TAC), a real-time polymerase chain reaction (RT-PCR) testing platform for 28 pathogens. Data were analyzed using descriptive statistics, and multivariate regression models of any TAC detections stratified by site and age.

**Data availability statement:** Data cannot be shared publicly without the express permission of the Liberian government per the approved protocol and government data sharing agreement. All data are owned by the National Public Health Institute of Liberia. A minimum, de-identified analytic dataset will be made available upon request following authorization from relevant authorities (DGHP GSLDS, askgslds@cdc.gov, +1 404 718 3120).

**Funding:** This project was funded by the United States Centers for Disease Control and Prevention (CDC) to the National Public Health Institute of Liberia through a cooperative agreement (NU2GGH001876) for the implementation of this surveillance project by the African Field Epidemiology Network (AFENET). The co-authors of the institutions receiving funding (JD, FT, LT, TN, MF) and implementing surveillance (EEQ, MAA, DA) acknowledge the funding. CDC as funders provided technical input on design, collection, analysis, and manuscript preparation. The opinions expressed herein are those of the authors and do not necessarily reflect the official position of the U.S. Centers for Disease Control and Prevention.

**Competing interests:** The authors have declared that no competing interests exists.

## Results

We enrolled 1506 AFI patients, 1206 (80%) from Redemption Hospital and 300 (20%) from Star of the Sea. AFI patients were predominantly female (69%) and had a median (interquartile range) age of 18 (7–27) years. Among the 699 (46%) that were TAC positive, 627 were detected from Redemption Hospital and 72 were detected from Star of the Sea Health Centre. Overall *Plasmodium* spp. (malaria) (96%) were the majority of detections followed by dengue virus (2%), *Streptococcus pneumoniae* (2%), and *Rickettsia* spp. (1%). We detected 19 co-infections [malaria co-infections (84%) being the most common]. Two pathogens with epidemic potential, *Neisseria meningitidis* (detected at Star of the Sea Health Centre) and Lassa virus (detected at Redemption Hospital), were also found. Patients with non-malaria TAC detections (n = 29) were higher at Star of the Sea Health Centre than Redemption Hospital (4% versus 1% respectively, $p < 0.05$). In multivariate regression for those ages 15 years and older at Redemption Hospital (adjusting for sex, age, pregnancy status, education, occupation, any medication use, measured fever at enrollment, headache, abdominal pain, vomiting/nausea and joint pain), any medication use (aOR=0.6, 95% CI = 0.4–0.9), measured fever at enrollment (aOR=3.5, 95% CI = 1.1–12.0), headache (aOR=1.7, 95% CI = 1.1–2.6 were statistically significant with any TAC detection. In multivariate regression for those ages 2–14 years at Redemption Hospital (adjusting for sex, age, abdominal pain, cough, vomiting/nausea, runny nose, and any animal exposure), having abdominal pain (aOR=1.9, 95% CI = 1.3–2.8), vomiting/nausea (aOR=0.6, 95% CI = 0.4–1.0), and any animal exposure (aOR=1.5, 95% CI = 1.0–2.3) were statistically significant with any TAC detection.

## Conclusion

This is the first laboratory evidence of dengue and rickettsial disease in humans in Liberia. Liberia's incipient AFI platform was successful exploring causes of fever in emerging infections and detected circulating pathogens beyond malaria. AFI surveillance data can assist in the prioritization of public health diagnostic and clinical capabilities to prevent, detect, and respond to emerging infectious disease threats in Liberia.

### Author summary

Among sick patients, fever is a very common symptom for many infectious diseases including those that have the potential to cause epidemics. In many low resource countries such as Liberia, the number of patients at healthcare facilities with fever is high and is often attributed to malaria but other bacterial, viral, or parasitic diseases may be circulating and are unknown. Therefore, exploring this possibility and identifying pathogens that may be the cause of fever is critical for public health. We report here our findings from testing febrile patients using a TaqMan Array Card designed to detect more than two dozen fever-causing

pathogens at two major health facilities in urban Monrovia, the capital city in Liberia. While just under half of febrile patients had any positive detections, more than 95% were due to malaria. However, we did detect pathogens causing meningitis and Lassa fever, diseases that have epidemic potential. We also detected dengue and Rickettsia, diseases that have never been reported in Liberia. Our surveillance data can be used to inform future febrile illness surveillance efforts and be used by Liberian health authorities for public health preparedness efforts to be able to detect and respond to infectious disease threats.

## Background

Acute febrile illness (AFI), whether in combination with localizing symptoms (*e.g.*, diarrhea, respiratory) or without localizing symptoms (*i.e.*, undifferentiated AFI), is a common manifestation of a variety of treatable and vaccine-preventable infectious diseases, as well as diseases of epidemic potential. In low- and middle-income countries, the burden of febrile illnesses is high, and fever is a leading cause among those seeking healthcare [1,2]. Determining the etiologies of these febrile illnesses often remains elusive [3]; with the exception of malaria tests, laboratory diagnostics may not be widely available due to testing resources availability and limited awareness of non-malaria febrile disease etiologies in low resource settings such as Liberia.

TaqMan Array Cards (TAC) have been designed and utilized previously for the surveillance of respiratory, diarrheal, and undifferentiated (i.e., non-respiratory, non-diarrheal) AFI illnesses [4–8]. TAC embeds quantitative real-time polymerase chain reaction (RT-PCR) assays for detecting multiple pathogens per patient specimen in a single card. This attribute makes multi-pathogen detection platforms such as TAC ideal for use where surveillance data on circulating pathogens may be lacking.

In 2013, the latest year of published data, 42% of all general medical consultations in Liberia were due to malaria [9]. In a country of 5.2 million, more than 1.6 million patients at Liberian health facilities were suspected of malaria (i.e., febrile) and more than 77,000 were hospitalized for severe malaria in 2021 [10]. Among those tested, however, malaria was confirmed in only 60%. Febrile illnesses such as malaria are a major driver of health consultations and place a substantial burden on the healthcare system in Liberia, yet malaria may not be the only etiologic cause. Establishing AFI surveillance at select health facilities with high numbers of febrile consultations can provide surveillance data for public health planning, laboratory diagnostics, and where appropriate, prevention of febrile illnesses.

The 2014–2015 Ebola epidemic in West Africa and the COVID-19 pandemic highlighted the need for greater investments in public health infrastructure to detect, monitor, and respond to disease and other threats to global health security [11,12]. This includes developing expanded capacities in laboratories and surveillance systems for infectious diseases that are already known to be circulating as well as those that may be newly emerging. Following the Ebola epidemic, national health authorities in Liberia (the National Public Health Institute of Liberia and Ministry of Health), together with partners, sought to improve surveillance capacity in the country for infectious diseases, particularly for those with epidemic potential. While the cause of the next epidemic may not be known beforehand, surveillance and understanding the etiologies of fever, a common symptom of infectious diseases, will help inform and prepare public health responses and develop appropriate capacities. Therefore, our objective in this AFI project is to explore causes of undifferentiated (i.e., non-respiratory and non-diarrheal) fever and their circulating pathogens at two sentinel sites in urban Monrovia, Liberia. We also characterize AFI patients detected to be positive for TAC pathogens and describe results from the implementation of AFI surveillance using TAC at two major health facilities.

## Methods

### Surveillance sites

Starting in December 2018, undifferentiated AFI surveillance was implemented in Monrovia, Montserrado County at two urban sentinel health facilities: Redemption Hospital and Star of the Sea Health Center. These public facilities were

chosen because of their high volume of patients, including Monrovia residents who may travel to/from different geographic regions within Liberia. Additionally, the communities surrounding these facilities comprise some of the largest informal settlements in Liberia are densely populated [13], and have water and sanitation and housing infrastructure needs: all conditions favoring the transmission of infectious diseases. Both health facilities were among the earliest to receive Ebola patients during the 2014–2015 epidemic in Liberia [14,15] making the sites ideal for sentinel AFI surveillance.

Redemption Hospital is a large 200 bed hospital that serves 90,000 people in New Kru Town and has an emergency unit, outpatient services (including antenatal care), maternal and pediatric wards, and an inpatient medical ward [16]. Redemption Hospital also receives patients from throughout Liberia who may be visitors to Monrovia due to its location near a major transit intersection and commercial hub. Star of the Sea is a primary health center which has no inpatient beds, but it is the only public medical facility that serves West Point, an informal settlement that had a projected 2014 population of nearly 35,000 [17] in a habitable area of just over 0.5 km². A major market is directly across from West Point with Star of the Sea Health Centre within 1 km. Both facilities provide healthcare services free of charge.

## Patient enrollment

During and for the years following the West Africa Ebola outbreak in 2014–2015, all patients were routinely screened and temporal temperatures taken upon entrance into health facilities or during triage for signs and symptoms of Ebola. These include fever, hemorrhagic symptoms, or contact with an Ebola-infected person or recent travel from an Ebola outbreak area (if applicable). AFI surveillance staff utilized this initial screening to further identify patients for project eligibility including retaking auxiliary temperature measurements and capturing other criteria on a standardized form (patients with temperatures of ≥ 37.5°C or a history of fever within the past week were considered eligible). Full inclusion and exclusion criteria are presented below. Every 10th eligible patient at Redemption Hospital was approached for consent and enrollment and every 5th eligible patient at Star of the Sea Health Centre was approached (due to lower patient volumes). Mondays and Wednesdays were selected as AFI surveillance days at Redemption Hospital with a target of 10 patients enrolled per day whereas patient enrollment at Star of the Sea Health Centre occurred on Mondays only with a target of five patients. These enrollment targets and days were selected to ensure the number TAC cards procured would last at least a full calendar year, Mondays were noted to have higher patient volumes, and Wednesday would allow adequate time for laboratory specimen processing. Following the determination of patient eligibility, patient informed consent was obtained (see below under Ethical Considerations) by AFI project staff.

## Inclusion and exclusion criteria

Inclusion criteria used for patient enrollment included being ≥ 2 years of age and either having a documented measured axillary, oral, or rectal temperature ≥ 37.5°C or presenting with a history or self-report of fever (i.e., "skin hot" or chills) within the previous 7 days.

Patients who did not consent/assent to participation or were incapable of providing consent/assent were excluded. Patients who previously enrolled in AFI surveillance within the past year or who were returning for the continuation of treatment of a known cause of fever were also excluded. Lastly, we excluded febrile patients with chief complaints that when examined by a healthcare provider specifically included injury or trauma, meningitis, soft tissue infection or cellulitis, urinary tract infection, septic arthritis, and other obvious sources of infections (e.g., bite wounds). Patient residency outside Monrovia was not an exclusion criteria as visitors may have been infected while in Monrovia or may import infectious illnesses.

## Specimen collection, storage, and transportation

Approximately 10 mL of venous whole blood was collected from AFI participants (5 mL from those 2–5 years) by venipuncture using Vacutainer syringes and sterile techniques with equal parts in an ethylenediaminetetraacetic acid (EDTA)

tube (*i.e.*, "purple top") and a serum tube (*i.e.*, "red top"). After collection, specimens were labeled with preprinted patient identification tags on barcoded cryogenic stickers and maintained in solar powered refrigerators between 2–8°C at the health facilities for a maximum of 72 hours. Motorcycle couriers transported specimens and maintained a monitored cold chain of 2–8°C to the National Public Health Reference Laboratory located at the Liberia Institute for Biomedical Research campus in Margibi County.

### Extraction, molecular testing, and run interpretation

Total nucleic acid material was extracted from 2 mL of whole blood specimen using Roche High Pure Viral Nucleic Acid Large Volume kits (Basel, Switzerland) following manufacturer protocol. During each extraction process, PhHV (Phocine Herpesvirus – $10^6$ copies) and MS2 (*Emesvirus zinderi*) bacteriophage ($10^7$ copies) were added to each of the samples to serve as controls for PCR inhibitory agents. A sample of nuclease-free water was extracted with each batch to monitor for contamination. A total of 25µL TaqMan Fast Virus 1-Step master mix (Thermo Fisher Scientific, Carlsbad, CA) was added to 75 µL of the total nucleic acid extract and loaded into each well of a TaqMan Array Card, centrifuged, and loaded onto a QuantStudio 7 Flex Real-Time PCR system (Thermo Fisher Scientific, Carlsbad, CA).

Interpretations of TAC run files were performed for each specimen using the QuantStudio Real–Time PCR software at a cycle threshold (Ct) cut-off value of 35.0 for positive samples. Quality controls were performed for each sample by checking for amplification below 35.0 in the MS2/PhHV wells and for no amplification in the blank control.

### TaqMan Array Cards

Customized TaqMan Array Cards were designed by the University of Virginia (UVA; Charlottesville, VA) with input from the US Centers for Disease Control and Prevention (CDC; Atlanta, GA) and manufactured by Applied BioSystems (Thermo Fisher Scientific, Carlsbad, CA). This multi-pathogen quantitative RT-PCR card contains assays to test for 12 bacterial, 12 viral, and 3 parasitic pathogens (Fig 1). Our TAC is nearly identical in design as previously described with assay performance characteristics described compared to individual PCR assays along with assay targets and sequences detailed [6]. Consideration of pathogens to include was based on those that may be a cause of fever among patients presenting to health facilities in West Africa and prioritizing those of public health importance. Specific targets included one for a general *Plasmodium* species (*Plasmodium* spp.) and species-specific *P. falciparum* and *P. vivax*. A specimen was considered to be malaria positive if any of the three *Plasmodium* TAC targets were positive. If TAC has multiple wells for a pathogen, a specimen was considered to be positive when having at least one positive for *Rickettsia* spp., *Leptospira* spp., *Neisseria meningitidis*, *Streptococcus pneumoniae*, or Lassa virus (Lassa fever). A more detailed description of primers and probes has been detailed previously [6].

### Data collection and analysis

Project personnel interviewed and collected demographic, clinical history, and risk factor information from enrolled patients on standardized paper forms. Malaria test results (i.e., microscopy, rapid diagnostic test) conducted as part of patient clinical care received at the facilities were also extracted from facility records. These forms along with interpreted TAC results were later entered electronically into EpiInfo (v.7.2.2.6) on a password protected laptop. Data analyses included descriptive statistics (i.e., frequencies, means, medians) for demographics, clinical characteristics, and risk factors. Chi-square statistical tests were calculated for proportions and t-tests for means. For any detection by TAC, stratified analysis was conducted for site and age group (2–14 and 15+years of age). Logistic regression was performed including variables for demographics (i.e., sex, age category), health facility, hospitalization status, any medication taken prior to enrollment (within the past 7 days before arrival or before specimen collection), measured fever (≥ 37.5°C) at enrollment, other

| Port | |
|---|---|
| **Left** | **Right** |
| *Bartonella* | *Bartonella* |
| *Brucella* spp. | *Brucella* spp. |
| *Burkholderia pseudomallei* (melioidosis) | Crimean-Congo hemorrhagic fever virus |
| *Coxiella burnetii* (Q Fever) | *Coxiella burnetii* (Q Fever) |
| Chikungunya virus | Dengue virus |
| *Orthoebolavirus bundibugyoense* (Bundibugyo Ebola virus) | *Orthoebolavirus sudanense* (Sudan Ebola virus) |
| Ebola NP (nucleoprotein) | Ebola VP40 (viral protein) |
| Hepatitis E virus | *Leishmania* and *Trypanosomiasis brucei* |
| *Leptospira* spp. | *Leptospira* spp. |
| Lassa virus | Lassa virus |
| *Orthomarburgvirus marburgense* (Marburg virus) and O'nyong-nyong virus | Nipah virus |
| *Neisseria menegitidis* | *Neisseria menegitidis* |
| MS2 (control) | phocine herpesvirus (PhHV, control) |
| 18S (control) | PhHV/MS2 (control) |
| *Orientia tsutsugamushi* | *Orientia tsutsugamushi* |
| *Plasmodium* spp. | Rift Valley fever virus |
| *Plasmodium falciparum* | *Plasmodium vivax* |
| *Rickettsia* spp. | *Rickettsia* spp. |
| *Salmonella enterica* | *Salmonella enterica* |
| *Salmonella* Typhi | 16S (control) |
| *Streptococcus pneumoniae* | *Streptococcus pneumoniae* |
| West Nile virus | Yellow fever virus |
| *Yersinia pesitis* | *Yersinia pesitis* |
| Zika virus (800) | Zika virus (1000) |

**Fig 1. Pathogen (TAC detections include pathogens causing bacterial (*Bartonella*, brucellosis, melioidosis, Q fever, leptospirosis, meningococcal disease, scrub typhus, rickettsia, salmonella, Typhoid fever, pneumococcus, plague), viral (CCHF, Ebola [general, Sudan, Bundibugyo], chikungunya, dengue, Hepatitis E, Lassa fever, Marburg, O'nyong-nyong, Nipah, West Nile, Yellow fever, Zika), and parasitic (leishmaniasis, trypanosomiasis, malaria [general, *P. falciparum*, *P. vivax*])) infections) layout of the customized TaqMan Array Card (TAC) used for acute febrile illness (AFI) surveillance, Liberia, December 2018–March 2020.**

symptoms (within the past 7 days), any animal exposure (based on animal type and type of contact within the past 30 days including bushmeat), and travel history for those ages 2–14 years and 15 years and older. For those ages 15 years and older, models also included highest education level achieved and occupation. Variables with $p < 0.1$ in bivariate analysis were included in the final multivariate model in addition to sex and age category.

## Ethical considerations

Following verification of eligibility, an AFI project background script was read in either English or Liberian pidgin English, questions addressed, and formal written informed consent (or assent for those <18 years of age) was obtained from patients or their parents/guardians. If a minor, parent, or guardian is illiterate, the relevant informed consent/assent form was verbally read to them by AFI staff in either English or Liberian Pidgin English in the presence of a witness. No compensation was provided for participation, but a small snack was provided to AFI surveillance participants.

This activity was reviewed and approved as non-research by Liberia's National Research Ethics Board and by the US Centers for Disease Control and Prevention (CGH 2018-337) and was conducted consistent with applicable US federal law and CDC policy.§§ See, e.g., 45 C.F.R. part 46.102(l)(2), 21 C.F.R. part 56; 42 U.S.C. §241(d); 5 U.S.C. §552a; 44 U.S.C. §3501 et seq.

## Results

From December 2018 through March 2020, a total of 1506 AFI patients were enrolled: 1206 (80%) from Redemption Hospital and 300 (20%) from Star of the Sea. Overall, AFI patients were predominantly female (69%), had a median (inter-quartile range) age of 18 (7–27) years with 42% under the age of 15 years, and completed primary school education or less (49% among those ≥ 15 years) (Table 1). Among enrolled adults, 13% were not currently working. AFI patients overwhelmingly resided within Montserrado County, with only six (6) (0.5%) from Redemption Hospital residing outside of Montserrado County. Among AFI patients enrolled at Redemption Hospital, 38% were hospitalized and 69% of reproductive age women (15–49 years) were pregnant.

At time of enrollment for Redemption Hospital and Star of the Sea Health Centre respectively, 98% and 24% ($p < 0.001$) of AFI patients had a measured fever with a mean of 3.8 (± 1.9) and 5.1 (± 3.6) ($p < 0.001$) days from fever onset to health facility visit. Besides fever, all but 24 AFI patients reported another symptom with the most common included headache (78%), abdominal pain (53%), and cough (52%) (Table 1); this was consistent among patients with any measured fever. For patients without measured fever, muscle pain (37%) was also commonly reported.

TAC testing detected at least one pathogen among 699 (46%) of patients with a greater proportion from Redemption Hospital than Star of the Sea (52% versus 24% respectively, $p < 0.05$). Overwhelmingly, *Plasmodium* spp. was the most commonly detected pathogen with the general *Plasmodium* spp. target positive for 96% of all detections (Table 2). Although the majority of malaria detections were positive for both *Plasmodium* spp. and *P. falciparum* (93%), 48 were positive only for *Plasmodium* spp. and no *P. vivax* was detected. Overall, *Plasmodium* spp. was also more commonly detected by TAC at Redemption Hospital than Star of the Sea Health Centre (51% versus 20% respectively, $p < 0.05$); this finding is consistent with malaria test results performed by facility (62% versus 12% respectively, $p < 0.05$) as part of clinical patient care. AFI patients with non-malaria TAC detections were higher at Star of the Sea Health Centre than Redemption Hospital (4% versus 1% respectively, $p < 0.05$). Among AFI patients who reported any travel history within the past 30 days, 16 (42%) had any TAC detections all for malaria. For the 6 (six) residing outside of Montserrado County, all 3 positive TAC detections were also for malaria. Other detected pathogens included dengue virus (n = 16; 2%), *Streptococcus pneumoniae* (n = 13; 2%), and Rickettsia (n = 10; 1%). Among the Rickettsia detections, three (3) cases reported having contact with dogs and/or bushmeat. Two (2) N*eisseria meningitidis* at Star of the Sea Health Centre and one (1) Lassa fever at Redemption Hospital were also detected. Patient characteristics for TAC detections of *Plasmodium* spp., dengue virus, *Rickettsia* spp. and *S. pneumoniae* are presented in Table 3.

Co-infections were detected among 19 (1%) AFI patients. Malaria accounted for 16 (84%) of these co-infections with malaria and dengue (n = 7; 37%) being the most common (Table 2). Four of the malaria co-infections tested positive for the general *Plasmodium* spp. target only. All but two (2) co-infections had a measured fever.

*Plasmodium* spp. were detected during all calendar months with the highest numbers in May through September 2019 (Fig 2) and with the proportions of *Plasmodium* spp. detections among AFI enrollments ranging between 56% to 69%. During these same months, only 1 dengue case was detected. Fig 3 presents non-malaria detections and positivity only.

In stratified analysis, multivariate regression models for any TAC detection among patients ages 15 years and older at Redemption Hospital, older age categories were statistically significant and protective to patients 15–19 years (Table 4) but not for other strata at Star of the Sea Health Centre and 2–14 years (Tables 5–7). Other statistically significant associations included pregnant patients 15 years and older at Redemption Hospital (aOR = 2.7, 95% CI: 1.7–4.4), any medication use (aOR = 0.6, 95% CI: 0.4–0.9), and having a measured fever at enrollment (aOR = 3.5, 95% CI: 1.1–12.0). Among patients 15 years and older, having headache (aOR = 1.7, 95% CI: 1.1–2.6) was significant among patients at Redemption Hospital and muscle pain at the Star of the Sea Health Centre (aOR = 3.2, 95% CI: 1.4–7.5) (Table 5). In multivariate regression models for any TAC detection among patients ages 2–14 years at Redemption Hospital, having abdominal

**Table 1. Demographics and clinical characteristics of enrolled patients presenting with acute febrile illness (AFI) by health facility, December 2018–March 2020.**

| | Total N (%) | Health Facility | |
| --- | --- | --- | --- |
| | | Redemption Hospital n (%) | Star of the Sea Health Centre n (%) |
| **Overall** | 1,506 | 1,206 | 300 |
| **In-patient** | 463 (31) | 463 (38) | 0 (0) |
| **Sex** | | | |
| Male | 472 (31) | 351 (29) | 121 (40) |
| Female | 1,034 (69) | 855 (71) | 179 (60) |
| **Pregnant[1]** | 393 (60) | 385 (69) | 8 (8) |
| **Age** | | | |
| Median (IQR) years | 18 (7–27) | 18 (7–26) | 20 (6–33) |
| **Age Category** | | | |
| 2–4 | 245 (16) | 183 (15) | 62 (21) |
| 5–9 | 228 (15) | 198 (16) | 30 (10) |
| 10–14 | 158 (10) | 129 (11) | 29 (10) |
| 15–19 | 221 (15) | 194 (16) | 27 (9) |
| 20–29 | 339 (23) | 277 (23) | 62 (21) |
| 30–39 | 151 (10) | 114 (9) | 37 (12) |
| 40–49 | 73 (5) | 47 (4) | 26 (9) |
| 50+ | 91 (6) | 64 (6) | 27 (9) |
| **Highest level of education[2]** | | | |
| No school | 309 (35) | 242 (35) | 67 (37) |
| Primary | 124 (14) | 84 (12) | 40 (22) |
| Secondary | 329 (38) | 272 (39) | 57 (32) |
| Higher | 113 (13) | 98 (14) | 15 (8) |
| **Occupation[2]** | | | |
| Missing | 90 (6) | 72 (6) | 18 (6) |
| Not working | 186 (12) | 139 (12) | 47 (16) |
| Student | 898 (60) | 729 (60) | 169 (56) |
| Professional | 219 (15) | 181 (15) | 38 (13) |
| Domestic services | 75 (5) | 58 (5) | 17 (6) |
| Manual labor | 33 (2) | 24 (2) | 9 (3) |
| **Any reported animal contact[3]** | 386 (26) | 326 (27) | 60 (20) |
| **Recent travel history** | 41 (3) | 20 (2) | 21 (7) |
| To another county | 38 (3) | 19 (2) | 19 (6) |
| To another country | 3 (<1) | 1 (<1) | 2 (1) |
| **Antibiotics or other medications taken before enrollment** | 515 (37) | 420 (37) | 95 (34) |
| **Measured fever at enrollment** | 1252 (83) | 1180 (98) | 72 (24) |
| **Other symptoms reported[4]** | | | |
| Headache | 1175 (78) | 976 (81) | 199 (66) |
| Abdominal Pain | 792 (53) | 740 (61) | 52 (17) |
| Cough | 786 (52) | 660 (55) | 126 (42) |
| Vomiting/Nausea | 364 (24) | 334 (28) | 30 (10) |
| Muscle Pain | 314 (21) | 200 (17) | 114 (38) |
| Runny Nose | 263 (17) | 240 (20) | 23 (8) |
| Arthralgia/Joint Pain | 217 (14) | 164 (14) | 53 (18) |

*(Continued)*

**Table 1.** (Continued)

| | Total N (%) | Health Facility | |
| --- | --- | --- | --- |
| | | Redemption Hospital n (%) | Star of the Sea Health Centre n (%) |
| Diarrhea | 201 (13) | 188 (16) | 13 (4) |
| **Malaria clinical test results[5]** | | | |
| Positive | 780 (52) | 744 (62) | 36 (12) |
| Negative | 535 (36) | 291 (24) | 244 (81) |
| No test done | 191 (13) | 171 (14) | 20 (7) |

[1]Among reproductive age (15–49 years) females; missing data include 100 (66 from Redemption Hospital, 34 from Star of the Sea Health Centre).

[2]Among ages 15 + years.

[3]Previous 30 days.

[4]Previous 7 days; other symptoms included rash (n = 14), jaundice (n = 3) and unexplained bleeding (n = 1).

[5]Malaria tests performed at health facility as part of clinical care.

pain (aOR = 1.9, 95% CI: 1.3–2.7) and vomiting/nausea (aOR = 0.6, 95% CI: 0.4–1.0) were also statistically significant. For AFI patients at Redemption Hospital, any animal exposure was significant for those 2–14 years (aOR = 1.5, 95% CI: 1.0–2.3) and nearly so for those ages 15 years and older (aOR = 1.3, 95% CI: 0.9–1.9) (Tables 4, 6). Interactions were not detected in multivariate models.

## Discussion

We report here our findings from AFI surveillance at two (2) sentinel health facilities in urban Monrovia, Liberia, using TAC, a multi-pathogen testing platform. Malaria overwhelmingly comprised the majority of detections by TAC. The U.S. President's Malaria Initiative has projected that over 2.2 million consultations for fever will occur in 2023 in Liberia [18]. While malaria constituted a majority of detections in the AFI project, it was not the only cause of fever; other detections via TAC included dengue virus, *Streptococcus pneumoniae*, and Rickettsia. As Liberian public health institutions together with health authorities strengthen and develop surveillance systems and response capabilities following the Ebola outbreak in 2014–2015 [19] and the COVID-19 pandemic, there is a need to identify other circulating pathogens beyond malaria [20].

To our knowledge, this surveillance platform provides the first laboratory evidence of human cases of dengue and Rickettsia in Liberia. Dengue has been suspected of circulating in Liberia given the presence of the *Aedes aegypti* vector and dengue reports in other West African countries [21,22]. Rickettsia had been detected previously in ticks sampled from domestic and wild animals in Liberia and neighboring Guinea [23]. For both dengue and Rickettsia positive detections, there was no evidence supporting importation as the only TAC detections for AFI patients among those reporting travel history or residence outside of Montserrado County were for malaria.

Surveillance data including from this AFI project can be utilized for public health preparedness and planning such as identifying diagnostic needs and developing related capacities. Of note, AFI surveillance also detected Lassa virus and *Neisseria meningitidis*; both have epidemic potential and are priority pathogens for Liberia's Integrated Disease Surveillance and Response (IDSR) platform [24]. For many of the pathogens, especially the high consequence ones (e.g., Ebola virus, Lassa virus, Yellow fever virus, etc.) on our TAC, one positive detection would constitute meeting an outbreak threshold. The detections of Lassa virus and *Neisseria meningitidis* underscore the value of AFI surveillance platforms for monitoring emerging high consequence pathogens as fever is a common presenting symptom.

Recognizing reference laboratory limitations and possible delays when shipping specimens outside of Liberia, ten (10) percent of AFI TAC cards were also reserved for emergency use by Liberian health authorities during suspected

**Table 2. Detected pathogens by TaqMan Array Card (TAC), AFI surveillance, Liberia, December 2018–March 2020.**

| | Total | Redemption Hospital | Star of the Sea Health Centre |
|---|---|---|---|
| | **N = 1,506** | **n = 1206** | **n = 300** |
| **Any TAC detection** | 699 (46) | 627 (52) | 72 (24) |
| **Malaria detected[1]** | | | |
| *Plasmodium* spp. | 670 (96) | 610 (97) | 60 (83) |
| *P. falciparum* | 622 (89) | 570 (91) | 52 (72) |
| *P. vivax* | 0 (0) | 0 (0) | 0 (0) |
| **Pathogens detected (non-malaria)[1]** | | | |
| Dengue | 16 (2) | 14 (2) | 2 (3) |
| *Streptococcus pneumoniae* | 13 (2) | 3 (0) | 10 (14) |
| *Rickettsia* spp. | 10 (1) | 7 (1) | 3 (4) |
| Hepatitis E | 2 (<1) | 2 (<1) | —— |
| *Neisseria meningitidis* | 2 (<1) | —— | 2 (3) |
| *Salmonella* spp. | 2 (<1) | 2 (<1) | —— |
| *Brucella* spp. | 1 (<1) | 1 (<1) | —— |
| Lassa fever | 1 (<1) | 1 (<1) | —— |
| *Leptospira* spp. | 1 (<1) | 1 (<1) | —— |
| **Co-detections[2]** | **N = 19** | **n = 14** | **n = 5** |
| Dengue, malaria | 7 (37) | 7 (50) | —— |
| *Rickettsia* spp., malaria | 3 (16) | 2 (14) | 1 (20)[3] |
| Hepatitis E, malaria | 2 (11) | 2 (14) | —— |
| *Neisseria meningitidis, Streptococcus pneumoniae* | 2 (11) | —— | 2 (40) |
| *Streptococcus pneumoniae,* malaria | 2 (11) | —— | 2 (40)[3] |
| Dengue, *Streptococcus pneumoniae* | 1 (5) | 1 (7) | —— |
| Lassa fever, malaria | 1 (5) | 1 (7)[3] | —— |
| *Brucella* spp., malaria | 1 (5) | 1 (7) | —— |

[1]Percentages calculated among the detections.

[2]Percentages calculated among the co-detections. Unless otherwise noted, malaria detections were TAC positive for both *Plasmodium* spp. and *P. falciparum* targets.

[3]Positive for *Plasmodium* spp. target only.

outbreaks of other priority pathogens. For example, TAC from this AFI surveillance project were used to test specimens during a suspected Lassa fever outbreak in March 2019, and TAC was previously used in a Liberian meningitis outbreak investigation in April 2017 [25]. While TAC is well-positioned for emergent use in outbreak settings, its cost may be prohibitive for routine use.

Although in relatively close geographic proximity (< 8 km), Redemption Hospital as a referral facility with hospital beds and Star of the Sea Health Centre as a lower-level primary care facility would attract different patients. Unsurprisingly, our data show differences in enrolled patient characteristics between Redemption Hospital and Star of the Sea. AFI enrollment at Redemption Hospital coincided with antenatal clinic days resulting in a higher enrollment of pregnant women at that facility. Redemption Hospital, as a referral hospital and near transit points, can attract patients from across Montserrado County and Liberia, while Star of the Sea functions as a primary health center that serves West Point residents. Our data show, however, that patients from Star of the Sea Health Centre proportionately reported more travel outside

**Table 3. Patient characteristics for select[1] detected pathogens by TaqMan Array Card (TAC), AFI surveillance, Liberia, December 2018–March 2020.**

| | TAC pathogen detected | | | |
| --- | --- | --- | --- | --- |
| | *Plasmodium* spp. | Dengue | *Rickettsia* spp. | *Streptococcus pneumoniae* |
| **Total** | 670 | 16 | 10 | 13 |
| **Site** | | | | |
| Redemption Hospital | 610 (91) | 14 (88) | 7 (70) | 3 (23) |
| Star of the Sea Health Centre | 60 (9) | 2 (13) | 3 (30) | 10 (77) |
| **Sex** | | | | |
| Male | 183 (26) | 7 (44) | 2 (20) | 9 (69) |
| Female | 516 (74) | 9 (56) | 8 (80) | 4 (31) |
| **Age category** | | | | |
| 2–4 | 79 (12) | 0 (0) | 2 (20) | 9 (69) |
| 5–9 | 92 (14) | 3 (19) | 0 (0) | 1 (8) |
| 10–14 | 81 (12) | 1 (6) | 2 (20) | 0 (0) |
| 15–19 | 154 (23) | 3 (19) | 1 (10) | 0 (0) |
| 20–29 | 170 (25) | 5 (31) | 2 (20) | 0 (0) |
| 30–39 | 58 (9) | 0 (0) | 2 (20) | 0 (0) |
| 40–49 | 18 (3) | 2 (13) | 0 (0) | 0 (0) |
| 50+ | 18 (3) | 2 (13) | 1 (10) | 3 (23) |
| **Pregnant[2]** | 273 (76) | 6 (75) | 1 (25) | 0 (0) |
| **Patient status** | | | | |
| In–patient | 225 (34) | 7 (44) | 3 (30) | 2 (15) |
| Out–patient | 445 (66) | 9 (56) | 7 (70) | 11 (85) |
| **Antibiotics or other medications taken before enrollment** | 215 (32) | 6 (38) | 4 (40) | 5 (38) |
| **Measured fever at enrollment** | 619 (92) | 14 (88) | 7 (70) | 6 (46) |
| **Other symptoms** | | | | |
| Headache | 556 (83) | 14 (88) | 7 (70) | 3 (23) |
| Abdominal Pain | 424 (63) | 10 (63) | 5 (50) | 2 (15) |
| Cough | 319 (48) | 10 (63) | 5 (50) | 11 (85) |
| Vomiting/Nausea | 470 (70) | 14 (88) | 7 (70) | 11 (85) |
| Muscle Pain | 134 (20) | 4 (25) | 3 (30) | 2 (15) |
| Runny Nose | 100 (15) | 1 (6) | 2 (20) | 2 (15) |
| Arthralgia/Joint Pain | 88 (13) | 2 (13) | 1 (10) | 0 (0) |
| Diarrhea | 102 (15) | 0 (0) | 2 (20) | 2 (15) |
| **Any animal exposure** | 215 (32) | 4 (25) | 3 (30) | 1 (8) |
| **Travel history** | 16 (2) | 0 (0) | 0 (0) | 0 (0) |

[1]Due to small numbers, only pathogens with 10 or more detections are presented.

[2]Among reproductive age (15–49 years) females; missing data include 31 for *Plasmodium* spp. and 1 for dengue.

of Montserrado County. A greater proportion of enrolled patients at Redemption Hospital had acute measured fever and had shorter times between symptom onset and seeking healthcare which may be the cause of more TAC detections at Redemption Hospital. Thus, this relative increase in detections could reflect differences in patients' health seeking behaviors for febrile illnesses, ease of access between the two hospitals, or patient differences in the surrounding communities.

Our AFI surveillance project can also help inform future surveillance efforts at these sentinel sites, among different age groups, and other similar settings. While the majority detections were for malaria and more so for Redemption Hospital than

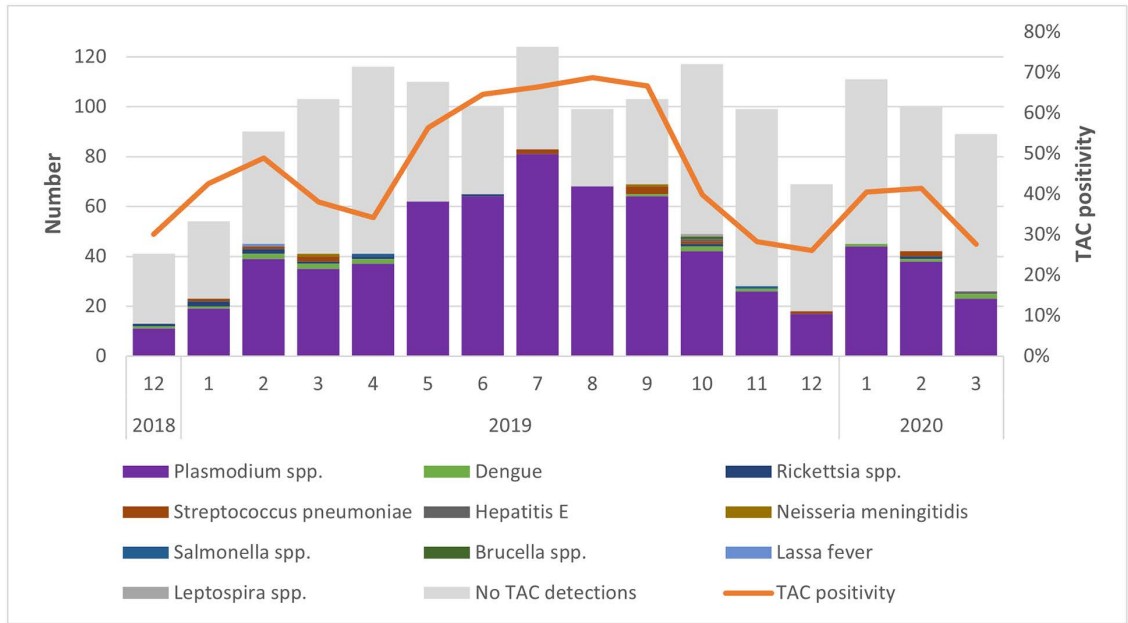

**Fig 2. TAC detected pathogens by year and month, AFI surveillance, Liberia, December 2018–March 2020.**

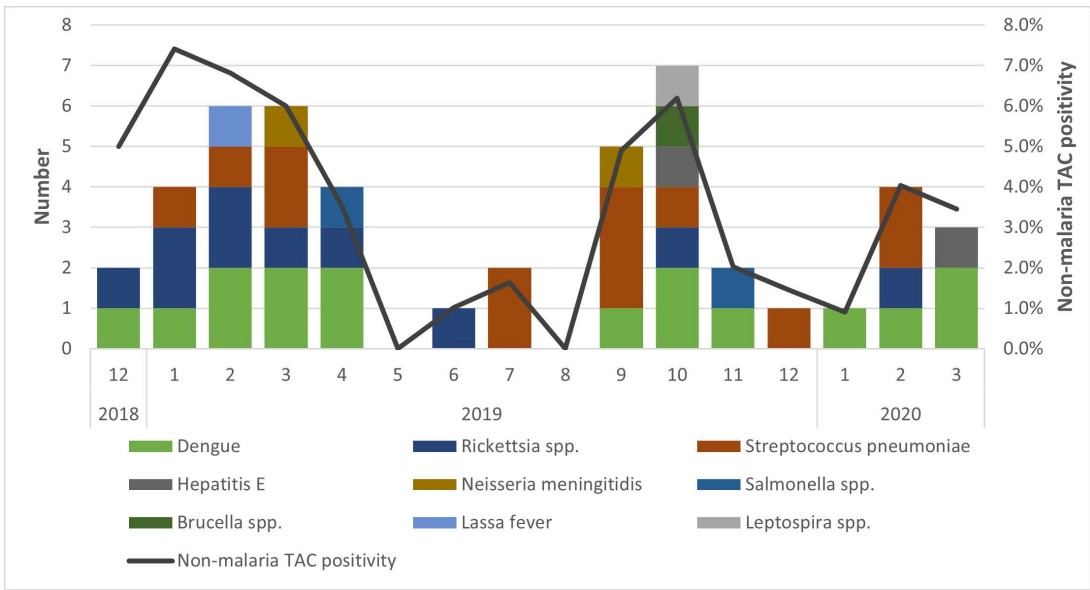

**Fig 3. Non-malaria TAC detected pathogens by year and month, AFI surveillance, Liberia, December 2018–March 2020.**

Star of the Sea Health Centre, our stratified multivariate analysis shows differences between measured fever and other symptoms that require further exploration. We also consider this initial AFI surveillance at these sentinel sites to be exploratory and aimed to be more inclusive and thus enrolled patients with self-reported fever, potential visitors to Monrovia, and 2 years and older. Future non-respiratory, non-diarrheal illness AFI surveillance efforts may want to consider refinement of enrollment criteria to increase detection yields including for other pathogens.

PLOS Neglected Tropical Diseases

**Table 4. Bivariate and multivariate regression model for any TAC pathogen detections among those ages 15 years and older, AFI surveillance, Redemption Hospital, Liberia, December 2018–March 2020.**

| | TAC positive | TAC negative | Odds Ratio (95% CI) | Adjusted Odds Ratio (95% CI) |
|---|---|---|---|---|
| | n (row %) | n (row %) | | |
| **Total** | 399 (57) | 297 (43) | —— | —— |
| **Sex** | | | | |
| Male | 46 (43) | 60 (57) | ref | ref |
| Female | 353 (60) | 237 (40) | 1.9 (1.3–3.0) | 0.6 (0.3–1.2) |
| **Age category** | | | | |
| 15–19 | 150 (77) | 44 (23) | ref | ref |
| 20–29 | 164 (59) | 113 (41) | 0.4 (0.3–0.6) | **0.4 (0.2–0.6)[1]** |
| 30–39 | 51 (45) | 63 (55) | 0.2 (0.1–0.4) | **0.3 (0.2–0.5)[1]** |
| 40–49 | 15 (32) | 32 (68) | 0.1 (0.1–0.3) | **0.3 (0.1–0.5)[1]** |
| 50+ | 19 (30) | 45 (70) | 0.1 (0.1–0.2) | **0.2 (0.1–0.5)[1]** |
| **Pregnant** | 274 (71) | 111 (29) | 3.5 (2.3–5.4) | **2.7 (1.7–4.4)[1]** |
| **Patient status** | | | | |
| In–patient | 166 (51) | 159 (49) | ref | ref |
| Out–patient | 233 (63) | 138 (37) | 1.6 (1.2–2.2) | 0.9 (0.6–1.3) |
| **Highest Level of Education** | | | | |
| No school | 123 (51) | 119 (49) | ref | ref |
| Primary | 54 (64) | 30 (36) | 1.7 (1.0–2.9) | 0.7 (0.4–1.4) |
| Secondary | 170 (63) | 102 (38) | **1.6 (1.1–2.3)[1]** | 0.8 (0.5–1.3) |
| Higher | 52 (53) | 46 (47) | 1.1 (0.7–1.8) | 1.0 (0.5–1.7) |
| **Occupation** | | | | |
| Not working | 36 (46) | 43 (54) | ref | ref |
| Student | 226 (64) | 125 (36) | **2.2 (1.3–3.5)[1]** | 1.4 (0.8–2.4) |
| Professional | 91 (50) | 90 (50) | 1.2 (0.7–2.1) | 1.1 (0.6–2.1) |
| Domestic services | 34 (59) | 24 (41) | 1.7 (0.9–3.4) | 1.2 (0.5–2.5) |
| Manual labor | 10 (42) | 14 (58) | 0.9 (0.3–2.2) | 1.1 (0.4–3.2) |
| Agriculture | 2 (67) | 1 (33) | 2.4 (0.2–27.4) | 3.5 (0.3–47.9) |
| **Antibiotics or other medications taken before enrollment** | 103 (43) | 139 (57) | **0.7 (0.6–0.9)[1]** | **0.6 (0.4–0.9)[1]** |
| **Measured fever at enrollment** | 394 (58) | 287 (42) | 2.7 (0.9–8.1) | **3.5 (1.1–12.0)[1]** |
| **Other symptoms** | | | | |
| Headache | 332 (59) | 229 (41) | 1.5 (1.0–2.1) | **1.7 (1.1–2.6)[1]** |
| Abdominal pain | 269 (61) | 170 (39) | **1.5 (1.1–2.1)[1]** | 1.3 (0.9–1.9) |
| Cough | 151 (56) | 120 (44) | 0.9 (0.7–1.2) | —— |
| Vomiting/Nausea | 288 (55) | 233 (45) | **0.7 (0.5–1.0)[1]** | 0.8 (0.5–1.2) |
| Muscle Pain | 92 (57) | 70 (43) | 1.0 (0.7–1.4) | —— |
| Runny Nose | 34 (53) | 30 (47) | 0.8 (0.5–1.4) | —— |
| Arthralgia/Joint Pain | 75 (51) | 73 (49) | **0.7 (0.5–1.0)[1]** | 0.9 (0.6–1.4) |
| Diarrhea | 51 (65) | 27 (35) | 1.5 (0.9–2.4) | —— |
| **Any animal exposure** | 116 (65) | 63 (35) | **1.5 (1.1–2.2)[1]** | 1.3 (0.9–1.9) |
| **Travel history** | 6 (46) | 7 (54) | 0.6 (0.2–1.9) | —— |

[1] $p < 0.05$.

**Table 5. Bivariate and multivariate regression model for any TAC pathogen detections among those ages 15 years and older, AFI surveillance, Star of the Sea Health Centre, Liberia, December 2018–March 2020.**

| | TAC positive | TAC negative | Odds Ratio (95% CI) | Adjusted Odds Ratio (95% CI) |
|---|---|---|---|---|
| | n (row %) | n (row %) | | |
| **Total** | 37 (21) | 142 (79) | —— | —— |
| **Sex** | | | | |
| Male | 12 (20) | 48 (80) | ref | ref |
| Female | 25 (21) | 94 (79) | 1.1 (0.5–2.3) | 0.9 (0.4–2.0) |
| **Age category** | | | | |
| 15–19 | 6 (22) | 21 (78) | ref | ref |
| 20–29 | 13 (21) | 49 (79) | 0.9 (0.3–2.8) | 0.8 (0.3–2.0) |
| 30–39 | 9 (24) | 28 (76) | 1.1 (0.3–3.7) | 0.9 (0.3–3.1) |
| 40–49 | 5 (19) | 21 (81) | 0.8 (0.2–3.1) | 0.6 (0.2–2.5) |
| 50+ | 4 (15) | 23 (85) | 0.6 (0.2–2.5) | 0.5 (0.1–2.2) |
| **Pregnant** | 3 (38) | 5 (63) | 2.0 (0.4–9.5) | —— |
| **Highest Level of Education** | | | | |
| No school | 16 (24) | 51 (76) | ref | —— |
| Primary | 5 (33) | 10 (67) | 0.5 (0.2–1.4) | |
| Secondary | 11 (19) | 46 (81) | 0.8 (0.3–1.8) | |
| Higher | 5 (33) | 10 (67) | 1.6 (0.5–5.4) | |
| **Occupation** | | | | |
| Not working | 7 (25) | 21 (75) | ref | —— |
| Student | 21 (25) | 64 (75) | 1.0 (0.4–2.6) | |
| Professional | 6 (16) | 32 (84) | 0.6 (0.2–1.9) | |
| Domestic services | 1 (6) | 16 (94) | 0.2 (0.0–1.7) | |
| Manual labor | 2 (22) | 7 (78) | 0.9 (0.1–5.1) | |
| Agriculture | 0 (0) | 2 (100) | —— | |
| **Antibiotics or other medications taken before enrollment** | 13 (21) | 48 (79) | 1.1 (0.5–2.3) | —— |
| **Measured fever at enrollment** | 5 (15) | 29 (85) | 0.6 (0.2–1.7) | —— |
| **Other symptoms** | | | | |
| Headache | 30 (21) | 112 (79) | 1.1 (0.4–2.9) | —— |
| Abdominal pain | 4 (13) | 28 (88) | 0.5 (0.2–1.5) | —— |
| Cough | 8 (14) | 48 (86) | 0.5 (0.2–1.3) | —— |
| Vomiting/Nausea | 34 (21) | 131 (79) | 1.0 (0.3–3.6) | —— |
| Muscle Pain | 28 (28) | 71 (72) | **3.1 (1.4–7.1)[1]** | **3.2 (1.4–7.5)[1]** |
| Runny Nose | 1 (13) | 7 (88) | 0.5 (0.1–4.5) | —— |
| Arthralgia/Joint Pain | 9 (19) | 39 (81) | 0.8 (0.4–2.0) | —— |
| Diarrhea | 0 (0) | 4 (100) | —— | —— |
| **Any animal exposure** | 15 (22) | 52 (78) | 1.2 (0.6–2.5) | —— |
| **Travel history** | 5 (33) | 10 (67) | 2.1 (0.7–6.5) | —— |

[1] $p < 0.05$.

Differences in the pathogens detected were noted between the two sentinel health facilities. Greater detections of *Streptococcus pneumoniae* and *Neisseria meningitidis* at Star of the Sea Health Centre may reflect more localized transmission of these pathogens within West Point. Our data also suggests that Star of the Sea functioned as a sentinel surveillance site for non-malaria detections. The distinction between pathogen detection by site designation may

**Table 6. Bivariate and multivariate regression model for any TAC pathogen detections among those ages 2 to 14 years, AFI surveillance, Redemption Hospital, Liberia, December 2018–March 2020.**

| | TAC positive | TAC negative | Odds Ratio (95% CI) | Adjusted Odds Ratio (95% CI) |
|---|---|---|---|---|
| | n (row %) | n (row %) | | |
| **Total** | 228 (45) | 282 (55) | —— | —— |
| **Sex** | | | | |
| Male | 108 (44) | 137 (56) | ref | ref |
| Female | 120 (45) | 145 (55) | 1.1 (0.7–1.5) | 1.0 (0.7–1.4) |
| **Age category** | | | | |
| 2–4 | 69 (38) | 114 (62) | ref | ref |
| 5–9 | 86 (43) | 112 (57) | 1.3 (0.8–1.9) | 1.1 (0.7–1.6) |
| 10–14 | 73 (57) | 56 (43) | **2.1 (1.4–3.4)[1]** | **1.7 (1.0–2.8)[1]** |
| **Patient status** | | | | |
| In–patient | 69 (50) | 69 (50) | ref | |
| Out–patient | 159 (43) | 213 (57) | 0.7 (0.5–1.1) | —— |
| **Antibiotics or other medications taken before enrollment** | 101 (48) | 109 (52) | 1.2 (0.9–1.8) | —— |
| **Measured fever at enrollment** | 225 (45) | 274 (55) | 2.2 (0.6–8.3) | —— |
| **Other symptoms** | | | | |
| Headache | 192 (46) | 223 (54) | 1.4 (0.9–2.2) | —— |
| Abdominal pain | 156 (52) | 145 (48) | **2.0 (1.4–2.9)[1]** | **1.9 (1.3–2.8)[1]** |
| Cough | 160 (41) | 229 (59) | **0.5 (0.4–0.8)[1]** | 0.7 (0.4–1.1) |
| Vomiting/Nausea | 144 (41) | 207 (59) | **0.6 (0.4–0.9)[1]** | **0.6 (0.4–1.0)[1]** |
| Muscle Pain | 17 (45) | 21 (55) | 1.0 (0.5–1.9) | —— |
| Runny Nose | 66 (38) | 110 (63) | 0.6 (0.4–0.9) | 0.8 (0.5–1.2) |
| Arthralgia/Joint Pain | 7 (44) | 9 (56) | 1.0 (0.4–2.6) | —— |
| Diarrhea | 52 (47) | 58 (53) | 1.1 (0.7–1.7) | —— |
| **Any animal exposure** | 79 (51) | 77 (49) | **1.4 (1.0–2.1)[1]** | **1.5 (1.0–2.3)[1]** |
| **Travel history** | 2 (33) | 4 (67) | 0.6 (0.1–3.4) | —— |

[1]$p < 0.05$.

be due to patient population differences, health seeking behavior differences, or disease ecology, and further exploration is needed which might include selection of other AFI sentinel sites for comparison. If an AFI platform is to be established in malaria endemic urban settings, our findings suggest that adolescents, commonly reported non-febrile symptoms (excluding vomiting/nausea), and animal exposures may be relevant variables to increase detection yields. Further exploration and analyses of non-malaria causes of fever in this setting should be undertaken. Similar to TAC used for AFI surveillance in Kenya, Tanzania, and other sub-Saharan African countries [26–28], TAC failed to detect any pathogens in over half of enrolled AFI patients. This could reflect pathogens not being tested for by this version of TAC, non-infectious causes of fever among enrolled patients, or delayed seeking of health services making PCR-based detections unlikely.

While relatively rare, co-infections were detected by TAC with malaria co-infections detected most often. These co-infections, as well as the detection of pathogens other than malaria, may have implications for patient care and treatment guidelines, particularly for febrile patients who may be unresponsive to malaria treatment. While the Liberia health facilities have the capacity to detect malaria infections, laboratory diagnostic capacity for other infectious causes of febrile illnesses for patient care may be more limited. It will be critical to prioritize laboratory resources based on available surveillance data, such as the data provided through this exploratory AFI project.

**Table 7. Bivariate and multivariate regression model for any TAC pathogen detections among those ages 2 to 14 years, AFI surveillance, Star of the Sea Health Centre, Liberia, December 2018–March 2020.**

| | TAC positive | TAC negative | Odds Ratio (95% CI) | Adjusted Odds Ratio (95% CI) |
|---|---|---|---|---|
| | n (row %) | n (row %) | | |
| **Total** | 35 (29) | 86 (71) | —— | —— |
| **Sex** | | | | |
| Male | 17 (28) | 44 (72) | ref | ref |
| Female | 18 (30) | 42 (70) | 1.1 (0.5–2.4) | 1.2 (0.5–2.9) |
| **Age category** | | | | |
| 2–4 | 18 (29) | 44 (71) | ref | ref |
| 5–9 | 8 (27) | 22 (73) | 0.9 (0.3–2.4) | 0.8 (0.3–2.3) |
| 10–14 | 9 (31) | 20 (69) | 1.1 (0.4–2.9) | 0.8 (0.3–2.4) |
| **Antibiotics or other medications taken before enrollment** | 9 (24) | 29 (76) | 0.7 (0.3–1.6) | —— |
| **Measured fever at enrollment** | 13 (34) | 25 (66) | 1.4 (0.6–3.3) | —— |
| **Other symptoms** | | | | |
| Headache | 20 (35) | 37 (65) | 1.8 (0.8–3.9) | —— |
| Abdominal pain | 8 (40) | 12 (60) | 1.8 (0.7–5.0) | —— |
| Cough | 16 (23) | 54 (77) | 0.5 (0.2–1.1) | 0.5 (0.2–1.1) |
| Vomiting/Nausea | 29 (28) | 76 (72) | 0.6 (0.2–1.9) | —— |
| Muscle Pain | 5 (33) | 10 (67) | 1.3 (0.4–4.0) | —— |
| Runny Nose | 3 (20) | 12 (80) | 0.6 (0.2–2.1) | —— |
| Arthralgia/Joint Pain | 0 (0) | 5 (100) | —— | —— |
| Diarrhea | 3 (33) | 6 (67) | 1.3 (0.3–5.3) | —— |
| **Any animal exposure** | 11 (30) | 26 (70) | 1.1 (0.5–2.5) | —— |
| **Travel history** | 3 (75) | 1 (25) | 8.0 (0.8–79.4) | 8.0 (0.8–83.0) |

Our analysis describes findings from the Liberia AFI platform up to early 2020 before the COVID-19 pandemic caused major disruptions to public health functions, patient care, and routine living within Liberia. COVID-19 testing of AFI patients was incorporated in April 2020. Then, in October 2021, the AFI surveillance platform later expanded to incorporate a tertiary referral hospital in Monrovia and two (2) hospitals in Nimba County. In December 2021, an updated version of TAC was implemented to include monkeypox (mpox) following an outbreak in 2017. This proved to be timely given the global outbreak of mpox in 2022 [29,30] and highlight the contributions to global health security that AFI surveillance can make.

## Limitations

Our analysis was subject to several limitations. Secondary confirmation was not possible for other pathogens detected by TAC due to in-country diagnostic capacities, low sample volume, or the specimen type being unsuitable for confirmation. With the high prevalence of malaria, we also could not definitively conclude TAC detections as the cause of fever illness especially for malaria detections given the possibility of asymptomatic infections. As TAC was not used for direct clinical care, we consider the positive detections as initial evidence deserving of further exploration. With TAC detections predominantly positive for malaria, there was insufficient data to thoroughly explore non-malaria TAC detections in our stratified multivariate analysis by site and age group. This too is another area for further exploration in future surveillance efforts. Whole blood also may not be an ideal specimen type or gold standard to detect some pathogens (e.g., blood culture for Typhoid fever). Although we attempted to enroll febrile patients during active infection, it may have been beyond a detectable time for PCR-based assays. Both situations would result in lower detections by TAC. Furthermore, while axillary temperature measurements are known to underestimate temperature readings versus rectal or oral measurements, we

do note that over 80% of patients had temperatures ≥ 37.5°C at enrollment and prior medication use including anti-pyretics could result in lower temperature measurements.

In addition, our testing capacity only sampled 25 patients per week and enrolled on select days among hundreds of febrile patients seeking healthcare daily at these two facilities. The enrolled AFI patients were not representative of the overall patient population as evident by high number of pregnant women enrolled for their ANC visits, but our objectives were to explore causes of non-respiratory and non-diarrheal fever and their circulating pathogens at two sentinel sites in urban Monrovia and to also characterize AFI patients with positive TAC detections. The non-representative sampling framework also limits the interpretation of our results, particularly for the demographics and multivariate models, as our results only reflect febrile patients who were enrolled and tested and broader generalizations would be inappropriate. While the high rates of malaria detections are consistent with other Liberia surveillance data, the proportions of other detected pathogens should be interpreted with caution and may not be reflective of the actual prevalence. Lastly, while TAC tests for a wide number of fever-causing pathogens, it is not inclusive of all that may be circulating.

## Conclusion

Liberia's AFI sentinel surveillance platform, using TAC, a multi-pathogen assay technology, has characterized possible etiologies of febrile illness at two sentinel health facilities in urban Monrovia. The primary objective of this AFI platform was to characterize the causes of fever followed by pathogen discovery; it detected non-malaria etiologies including priority diseases with epidemic potential. Further exploration of non-malaria causes of fever should be undertaken in this setting. TAC also performed well as an adjunct to Liberia's national reference laboratory platform – serving as an exploratory and potential outbreak identification tool in cases with unknown etiology, particularly for high consequence pathogens, when the standard reference platform was resource-challenged. These surveillance data can be used by Liberian health authorities to inform priorities for laboratory diagnostic and clinical treatment resources and algorithms and surveillance strategies. We urge continued investments and development in Liberia's capacity to prevent, detect, and respond to emerging infectious diseases.

## Acknowledgments

In addition to the patients who participated in this AFI surveillance project, the authors would like to extend their gratitude and acknowledge the following individuals and organizations for their many contributions.

*Redemption Hospital*: Hawa Robinson, Quoisey Korzu, Othello N. Toe, Tamba Borbor

*Star of the Sea Health Centre*: Maimah Smith, John Suah, Joseph Gbato and Authur Cole

*National Reference Laboratory*: Carmilia Johnson, Aaron Momolu

*University of Virginia*: Professor Eric Houpt Laboratory

*Riders for Health*: Joseph Mehdeh, Trunos Grison, Marcus Kolle, Alexander Smith, Emmanuel Fahnbulleh

*National Public Health Institute Liberia:* Tolbert Nyenswah, Thelma V. Nelson, Trokon Yeabah

*Ministry of Health:* Gborbee Logan, Catherine Cooper, Yatta Wapoe

*Centers for Disease Control and Prevention:* Olga Henao, Joel Montgomery, Stefano Rosillo, Madeline Farron, Jolie Dennis, Christine Dowie, Nicolas Schaad

*World Health Organization*: Mesfin Zbelo

The findings and conclusions in this report are those of the author(s) and do not necessarily represent the official position of the U.S. Centers for Disease Control and Prevention.

## Author contributions

**Conceptualization:** Terrence Q. Lo, Elijah Paa Edu-Quansah, John Dogba, Fahn Taweh, Lekilay Tehmeh, Thomas Nagbe, Paul Whesseh, Eric Houpt, Jie Liu, Maame Amo-Addae, Davis Ashaba, Victoria Katawera, Denise Roth Allen, Barry Fields, Mosoka Fallah, Desmond Williams.

**Data curation:** Elijah Paa Edu-Quansah, John Dogba, Fahn Taweh, Jie Liu, Darwin J. Operario, Davis Ashaba, Daniel W. Martin, Amanda Balish, Barry Fields, Gulu Gwesa.

**Formal analysis:** Terrence Q. Lo, Elijah Paa Edu-Quansah, John Dogba, Fahn Taweh, Eric Houpt, Jie Liu, Darwin J. Operario, Davis Ashaba, Daniel W. Martin, Amanda Balish, Barry Fields, Desmond Williams.

**Funding acquisition:** Maame Amo-Addae, Barry Fields, Mosoka Fallah, Desmond Williams.

**Investigation:** Terrence Q. Lo, Elijah Paa Edu-Quansah, John Dogba, Fahn Taweh, Lekilay Tehmeh, Thomas Nagbe, Daniel W. Martin, Denise Roth Allen.

**Methodology:** Terrence Q. Lo, Elijah Paa Edu-Quansah, John Dogba, Fahn Taweh, Lekilay Tehmeh, Thomas Nagbe, Paul Whesseh, Dore Diabe, Eric Houpt, Jie Liu, Darwin J. Operario, Maame Amo-Addae, Davis Ashaba, Victoria Katawera, Daniel W. Martin, Denise Roth Allen, Amanda Balish, Barry Fields, Gulu Gwesa, Mosoka Fallah, Desmond Williams.

**Project administration:** Elijah Paa Edu-Quansah, John Dogba, Fahn Taweh, Lekilay Tehmeh, Paul Whesseh, Dore Diabe, Maame Amo-Addae, Davis Ashaba, Denise Roth Allen, Barry Fields, Mosoka Fallah, Desmond Williams.

**Resources:** John Dogba, Fahn Taweh, Eric Houpt, Darwin J. Operario, Maame Amo-Addae, Davis Ashaba, Amanda Balish, Barry Fields.

**Software:** Jie Liu, Darwin J. Operario.

**Supervision:** Terrence Q. Lo, Elijah Paa Edu-Quansah, John Dogba, Paul Whesseh, Dore Diabe, Eric Houpt, Jie Liu, Maame Amo-Addae, Davis Ashaba, Denise Roth Allen, Barry Fields, Mosoka Fallah, Desmond Williams.

**Validation:** Terrence Q. Lo, Elijah Paa Edu-Quansah, John Dogba, Fahn Taweh, Lekilay Tehmeh, Jie Liu, Darwin J. Operario, Amanda Balish, Gulu Gwesa.

**Writing – original draft:** Terrence Q. Lo, Elijah Paa Edu-Quansah.

**Writing – review & editing:** Terrence Q. Lo, Elijah Paa Edu-Quansah, John Dogba, Fahn Taweh, Lekilay Tehmeh, Thomas Nagbe, Paul Whesseh, Eric Houpt, Jie Liu, Darwin J. Operario, Victoria Katawera, Daniel W. Martin, Denise Roth Allen, Amanda Balish, Barry Fields, Gulu Gwesa, Mosoka Fallah, Desmond Williams.

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
