## [Decision Letter · Decision Letter 0]

17 Dec 2024

Acute Febrile Illness Surveillance Using TaqMan Array Cards in Two Urban Health Facilities, Monrovia, Liberia, December 2018 – March 2020

Dear Dr. Lo,

Thank you for submitting your manuscript to PLOS Neglected Tropical Diseases. After careful consideration, we feel that it has merit but does not fully meet PLOS Neglected Tropical Diseases's publication criteria as it currently stands. Therefore, we invite you to submit a revised version of the manuscript that addresses the points raised during the review process.

Please submit your revised manuscript within 60 days Feb 15 2025 11:59PM. If you will need more time than this to complete your revisions, please reply to this message or contact the journal office at plosntds@plos.org. Please include the following items when submitting your revised manuscript:

We look forward to receiving your revised manuscript.

Kind regards,

Eric HY Lau, Ph.D.

Academic Editor

Mabel Carabali

Section Editor

Shaden Kamhawi

co-Editor-in-Chief

Paul Brindley

co-Editor-in-Chief

**Additional Editor Comments :**

The Authors are expected to address all the criticisms by all Reviewers. In particular, please clarify the study aim, assess the impact of the TaqMan array card test performance on the results (Reviewers #1 and #2), describe the target population, consider the impact of imported infections, and assess the impact of fever on the case definition and related analyses (Reviewer #2). In additional to these comments, please address:

1. Table 1. Please indicate the numbers of inpatients and outpatients

2. The analyses presented in Tables 3 and 4 are pooled data from Redemption Hospital and Star of the Sea Health Centre. However, the target patients and disease severity between the hospital and health center, and likely the risk factors are different (as shown in Table 1). A stratified analysis by health facilities should be better performed.

3. Tables 3 & 4. Please present the row percentages (TAC positive as the main outcome)

4. Tables 3 & 4. Please explain why some estimates (e.g. animal exposure, travel history) were not presented.

5. Figure 2, the colours for Plasmodium spp and Lassa fever were too similar.

**Journal Requirements:**

At this stage, the following Authors/Authors require contributions: Terrence Q. Lo, Elijah Paa Edu-Quansah, John Dogba, Fahn Taweh, Lekilay Tehmeh, Thomas Nagbe, Paul Whesseh, Dore Diabe, Eric Houpt, Jie Liu, Darwin Operario, Maame Amo-Addae, Davis Ashaba, Victoria Katawera, Daniel Martin, Denise Allen, Amanda Balish, Barry Fields, Gulu Gwesa, Mosoka Fallah, and Desmond Williams. Please ensure that the full contributions of each author are acknowledged in the "Add/Edit/Remove Authors" section of our submission form.

3) We noticed that you used the phrase 'unpublished ' in the manuscript. We do not allow these references, as the PLOS data access policy requires that all data be either published with the manuscript or made available in a publicly accessible database. Please amend the supplementary material to include the referenced data or remove the references.

4) Thank you for including an Ethics Statement for your study. Please include:

i) The approval number(s), or a statement that approval was granted by the named board(s)

ii) A statement that formal consent was obtained (must state whether verbal/written) from the parent/guardian for child participants.

5) Please upload all main figures as separate Figure files in .tif or .eps format. For more information about how to convert and format your figure files please see our guidelines:

6) Please ensure that all Figure and Table files have corresponding citations and legends within the manuscript. Currently, Figure 3 and Table 4 in your submission file inventory do not have in-text citations. Please include the in-text citations of the figure and the table.

7) We have noticed that you have uploaded Supporting Information files, but you have not included a list of legends. Please add a full list of legends for your Supporting Information files after the references list.

8) Please remove the supplementary table from the main figures and tables file and upload it with the file type 'Supporting Information'. Please ensure that each Supporting Information file has a legend listed in the manuscript after the references list.

9) In the online submission form, you indicated that  "A minimum, de-identified analytic dataset will be made available upon request." All PLOS journals now require all data underlying the findings described in their manuscript to be freely available to other researchers, either

1. In a public repository

2. Within the manuscript itself

3. Uploaded as supplementary information.

10) Please amend your detailed Financial Disclosure statement. This is published with the article. It must therefore be completed in full sentences and contain the exact wording you wish to be published.

11) Please ensure that the funders and grant numbers match between the Financial Disclosure field and the Funding Information tab in your submission form. Note that the funders must be provided in the same order in both places as well. Please include the grant number in the Financial Disclosure statement.

Please indicate by return email the full and correct funding information for your study and confirm the order in which funding contributions should appear. Please be sure to indicate whether the funders played any role in the study design, data collection and analysis, decision to publish, or preparation of the manuscript.

**Reviewers' Comments:**

Reviewer's Responses to Questions

**Key Review Criteria Required for Acceptance?**

**Methods**

-Are the objectives of the study clearly articulated with a clear testable hypothesis stated?

-Is the study design appropriate to address the stated objectives?

-Is the population clearly described and appropriate for the hypothesis being tested?

-Is the sample size sufficient to ensure adequate power to address the hypothesis being tested?

-Were correct statistical analysis used to support conclusions?

-Are there concerns about ethical or regulatory requirements being met?

Reviewer #1: 1. You conducted a multivariate model to assess risk factors for any Tac +ve result. There wasn't a study aim that aligned with this analysis, and I think it would be helpful to describe the rationale for this analysis - presumably given the positive results were overwhelmingly malaria, essentially you are presenting risk factors for a positive falciparum PCR assay.

2. I recommend explicitly stating the pathogens tested for using the Tac card - I don't think the assay names in the appendix (eg "PhHV" or "Sudan" or "MS") are really sufficient.

Reviewer #2: Lines 107 – 108: The population of the study is insufficiently described. Patients attending the two health care facilities were seen from a “variety” of demographic and geographic regions from Liberia is too vague. And why highlighting this variety whereas the aim of the study was to explore causes of fever and circulating pathogens in urban Monrovia?

This sentence suggests also that these two health care facilities have a spatial representativeness of the population in Liberia. Geographical provenance of patients should be described more precisely for the two health care facilities, in particular their attractivity area.

Lines 114 - 115: “… a projected 2014 population of nearly 35000 in an area of just over 0.5 km²” gives 70000 inhabitants/km² that seems very high. Please check both population and area.

Line 117: At patient enrollment, was a self-questionnaire used or was it self-administered? Please, clarify this point.

Line 120: Please, could you recall symptoms of Ebola and what happened when a patient had these. Was it an exclusion criterion? However, this pathogen is searched with TAC. I don’t understand.

Lines 123 – 127: During the pseudo-random sampling some days of the week were chosen and other were not, why?

Line 135: What is exactly a self-report of fever? Feeling hot? Chills reported within the previous 7 days.

Line 135: There were no spatial criteria of exclusion. Did you mean that patients coming from outside Liberia were also included?

Line 138: “… a documented measured axillary, oral, or rectal temperature ≥ 37.5°C…” is one of the inclusion criteria. It is admitted that an axillary measure underestimate temperature of 0.5°C versus rectal measure. Please mention this bias of measurement.

Lines 137 – 139: In my opinion, the case definition is very sensitive and/or with some level of uncertainty as fever could be declared. I think that TAC negative tests could be also done wrongly for patients with no fever. The case definition itself could explain also the high percentage of negative TAC tests discussed lines 320 - 323, in particular at the Star of the Sea health care centre.

Line 141: “Exclusion criteria included…” is a little bit confusing.

Line 145: “… with a clear source of infection”. I don’t understand well what does it mean? Patients with no clinical cause of fever identified despite a diagnostic approach based on clinical signs and/or paraclinical tests could be an additional criterion of the case definition? At this point, the reader may wonder about the objective of the study which was initially to explore the causes of fever and the pathogens circulating in the urban area of Monrovia. It is necessary to clarify the objective of the study and the causes of fever that were explored.

**Results**

-Does the analysis presented match the analysis plan?

-Are the results clearly and completely presented?

-Are the figures (Tables, Images) of sufficient quality for clarity?

Reviewer #1: 1. Do you think the +ve falciparum PCR assays in the Tac card represent the cause of illness? Presumably with such a high proportion positive, many will have asymptomatic infection. What was the correlation with more conventional malaria tests?

2. Given the high proportion with negative results from the Tac card, were there any other conventional diagnostic tests performed (eg blood or urine culture), and are these results available?

Reviewer #2: Line 217: Please, inform the percentage of patients > 15 years.

Lines 221 – 222: What were the causes of admissions at the Redemption hospital?

Lines 240 – 244: For the other pathogens detected than Plasmodium, were there imported cases?

Line 258: In table 3, I am not sure for the OR of sex as the sum of TAC negative patients according to gender is more than 439. Please check for this.

Tables and figures

Figure 1

Figure 1 seems to be a screen capture. Some pathogens have abbreviations, MS2, 18S, CCHF, PhHV, 16S, and should be informed. Please, could you improve this figure and render it clearer.

Some pathogens seem not detected by TAC as antimicrobial resistance, tuberculosis, Borrelia, mycobacteria, influenza, Q fever, VRS, adenovirus. The epidemiological relevance of these last in the AFI surveillance should be discussed.

It would be interesting in knowing the distribution of main pathogens (Plasmodium, Salmonella, Streptococcus…) among age groups (< and ≥ 15 years).

Table 1

As pregnant exhibit 45 and 4.4% of female according to health care facilities, why not include this variable in the univariate analysis.

For missing and unknown answers, how did they analyse?

Table 4

Abdominal pain is significant for positive TAC tests whereas cough is significant for negative one. Does it mean that TAC could potentially detect more pathogens with a digestive tropism than pathogens with a respiratory one?

**Conclusions**

-Are the conclusions supported by the data presented?

-Are the limitations of analysis clearly described?

-Do the authors discuss how these data can be helpful to advance our understanding of the topic under study?

-Is public health relevance addressed?

Reviewer #1: 1. Among limitations, it is also worth noting that PCR assays are not the most sensitive test for many pathogens. For example, blood culture using automated systems is the reference test for bacteremia (eg typhoid) and diseases such as leptospirosis are complicated with different sample types (eg urine) or assays (eg MAT serology) preferred later in the illness.

2. There are major differences in the proportion with a TAC detection between sites - could you discuss this in more detail? Might it relate to severity of illness and/or prevalence of respiratory viral infection?

Reviewer #2: Line 282: If appropriate, dengue cases should be discussed according to imported cases Cf. my comments for lines 135 and 240 – 244.

Lines 295 – 296: Efficiency of TAC could be discussed with that of some RDT (malaria, meningitis…). Also, what is the diagnostic value of TAC (sensitivity, specificity…) according to the main pathogens encountered in Liberia? Is fever the best sign to prescribe a TAC test versus other symptoms or combination of symptoms like influenza-like illness in example.

Lines 313 - 316: The distinction between pathogen detection by site should be discussed also regarding a selection bias as the recruitment of an hospital is different of that of a health care centre.

Line 318: Animal exposure is searched in the previous 30 days and is significant for positive TAC. Which pathogens were expected to be detected? How this variable was measured? The (self-?) administered questionnaire would be interesting in supplementary material to see how questions were formulated. I think a memory bias should be also discussed for this variable.

Line 321: Please, see my comments for lines 137 – 139 and line 145.

Lines 346 – 348: You can add that another cause of non-representativity of the overall population is the pseudo-random sampling as pointed in lines 123 – 127. As the aim of the study was to explore causes of fever and circulating pathogens in urban Monrovia, it is more a spatial representativity which should be reached and discussed here.

**Editorial and Data Presentation Modifications?**

Reviewer #1: nil

Reviewer #2: (No Response)

**Summary and General Comments**

Reviewer #1: nil additional

Reviewer #2: The study population should be better described as it concerned inhabitants in Monrovia. The case definition seems very sensitive and need additional explanations and discussion.

Introduction

Lines 71 – 73: I don’t agree with the fact that complex and expensive laboratory diagnostics are not widely available as there are a large offer of rapid diagnostic tests even if their efficiency need more study, cf. Reference Bouzid D et al. Rapid diagnostic tests for infectious diseases in the emergency department. Clin Microbiol Infect. 2021 Feb;27(2):182-191. doi: 10.1016/j.cmi.2020.02.024. Please could you discuss the role of TDRs in the epidemiological surveillance and link their use to that of TAC?

Lines 82 – 84: Could you quantify the impact of febrile illnesses on health consultations and hospital admissions at Monrovia/Liberia instead of “substantial burden” that remains vague.

Line 99: “… implementation of AFI sentinel surveillance using TAC at two health”, the sentence is not finished.

PLOS authors have the option to publish the peer review history of their article (what does this mean? ). If published, this will include your full peer review and any attached files.

**Do you want your identity to be public for this peer review?** For information about this choice, including consent withdrawal, please see our Privacy Policy .

Reviewer #1: No

Reviewer #2: **Yes:** JUSOT

**Figure resubmission:**

**Reproducibility:**



---

## [Decision Letter · Decision Letter 1]

9 Oct 2025

Response to Reviewers
Revised Manuscript with Track Changes
Manuscript

Shaden Kamhawi

co-Editor-in-Chief

Paul Brindley

co-Editor-in-Chief

**Additional Editor Comments:**

**Journal Requirements:**

1) Please provide an Author Summary. This should appear in your manuscript between the Abstract (if applicable) and the Introduction, and should be 150-200 words long. The aim should be to make your findings accessible to a wide audience that includes both scientists and non-scientists. Sample summaries can be found on our website under Submission Guidelines:

2) Please upload a copy of Figure Figure 1 which you refer to in your text on page 11.. Or, if the figure is no longer to be included as part of the submission please remove all reference to it within the text.

3) Tables should not be uploaded as individual files. Please remove these files and include the Tables in your manuscript file as editable, cell-based objects. For more information about how to format tables, see our guidelines:

https://journals.plos.org/plosntds/s/tables

**Reviewers' comments:**

**Key Review Criteria Required for Acceptance?**

**Methods:**

-Are the objectives of the study clearly articulated with a clear testable hypothesis stated?

-Is the study design appropriate to address the stated objectives?

-Is the population clearly described and appropriate for the hypothesis being tested?

-Is the sample size sufficient to ensure adequate power to address the hypothesis being tested?

-Were correct statistical analysis used to support conclusions?

-Are there concerns about ethical or regulatory requirements being met?

Reviewer #1: Nil - previous suggestions have been addressed.

Reviewer #2: It is a descriptive study based on a sentinel surveillance with two sites sufficiently depicted. This design allows to answer to the main aim of the study, to explore causes of undifferentiated fever and their causal infectious agents. Populations attending the two sentinel sites are sufficiently described.

For reason of limited resources, the authors used a pseudo-random sample from the two sentinel sites. As they performed a multivariate logistic regression after stratification on age and sentinel site, they have to discuss the probably insufficient statistical power of the study in limitations, in discussion.

To be more precise, did in-patient already hospitalized at their inclusion or were they admitted after retaking temperature upon entrance into health facilities?

The pragraph about ethical considerations is clear.

**Results:**

-Does the analysis presented match the analysis plan?

-Are the results clearly and completely presented?

-Are the figures (Tables, Images) of sufficient quality for clarity?

Reviewer #1: nil - previously addressed.

Reviewer #2: The analysis match the analysis plan and the results are complete.

**Conclusions:**

-Are the conclusions supported by the data presented?

-Are the limitations of analysis clearly described?

-Do the authors discuss how these data can be helpful to advance our understanding of the topic under study?

-Is public health relevance addressed?

Reviewer #1: In the limitations section:

Line 398-399: "Our data also suggests that Star of the Sea functioned sentinel surveillance site for non-malaria detections." I think this sentence needs revision for clarity. Perhaps '...functioned as a sentinel..."

Line 438-439 "Utilizing whole blood also may not be an ideal specimen type to detect some pathogens (e.g. blood culture for Typhoid fever)" Are you suggesting bone marrow aspirate is the preferred sample type, or that PCR is less sensitive than culture?

In your conclusions:

I appreciate that you identify that you have insufficient non-malarial TAC positive participants, but it would be useful to flag that an analysis to identify those at high risk of testing positive for non-malarial pathogens would be helpful. Given the high cost of TAC-array cards, it would be ideal to know which group of patients to deploy them in.

Reviewer #2: In the conclusion, this seems too far going writing performing TAC can serve as an outbreak indentification tool in the article whereas it seems more a warning tool. Indeed, a pseudo-random sampling is not sufficiently exhaustive to make currently outbreak detection with sentinel site surveillance. A complete surveillance system must have a clear and acurate case definition for each infectious disease to reach this goal and the alone criteria based on fever is too restrictive. I leave the authors explaining the positioning of TAC.

For discussion, i remind to discuss the probably insufficient statistical power of the study in limitations.

In limitations, the authors wrote (line 439) "Although we attempted to enroll febrile patients during active infection". As it is depicted in the paragraph Patient enrollment, it seems difficult to catch infection at active phase on the alone fever criteria.

Even though it is not part of the objectives, it would be interesting in having some cost data relating to TAC.

The remaining of the conclusion is relevant for public health.

**Editorial and Data Presentation Modifications?**

Reviewer #1: nil

Reviewer #2: Minor modification

Line 289: the majority instead of themajority

**Summary and General Comments:**

Reviewer #1: Thank you for the opportunity to review your revised manuscript. Overall, my comments/queries have been addressed. I have a few remaining/new minor comments (listed above)

Reviewer #2: It is an interesting study about the difficult questioning of the "unobvious" causes of febrile illness. It is based on a sentinel surveillance with two sites that differ in their population coverage. In this way, there is potentially a large variety of causal agents. The study asks the place of tests like TAC in warning, surveillance and their cost-effective yield.

The limited resources led to use a pseudo-random sampling that questioned the representativness and the statistical power of the study whereas a multivariate logistic regression is performed.

The discussion is complete and conclusion is relevant for the challenge in detection of emergent and re-emergent infectious disease.

PLOS authors have the option to publish the peer review history of their article (what does this mean? ). If published, this will include your full peer review and any attached files.

**Do you want your identity to be public for this peer review?** For information about this choice, including consent withdrawal, please see our Privacy Policy .

Reviewer #1: No

Reviewer #2: **Yes:** Jean-François JUSOT

**Figure resubmission:**
---

## [Editor Report · Decision Letter 2]

20 Jan 2026

Dear Dr. Lo,

We are pleased to inform you that your manuscript 'Acute Febrile Illness Surveillance Using TaqMan Array Cards in Two Urban Health Facilities, Monrovia, Liberia, December 2018 – March 2020' has been provisionally accepted for publication in PLOS Neglected Tropical Diseases.

Best regards,

Eric HY Lau, Ph.D.

Academic Editor

Mabel Carabali

Section Editor

Shaden Kamhawi

co-Editor-in-Chief

Paul Brindley

co-Editor-in-Chief

Thanks for addressing all the editor’s and reviewers' comments. Congratulations on the excellent work!

---

## [Editor Report · Acceptance letter]

Dear Dr. Lo,

We are delighted to inform you that your manuscript, "Acute Febrile Illness Surveillance Using TaqMan Array Cards in Two Urban Health Facilities, Monrovia, Liberia, December 2018 – March 2020," has been formally accepted for publication in PLOS Neglected Tropical Diseases.

Best regards,

Shaden Kamhawi

co-Editor-in-Chief

Paul Brindley

co-Editor-in-Chief
